# Identification and Characterization of Downy Mildew-Responsive microRNAs in Indian *Vitis vinifera* by High-Throughput Sequencing

**DOI:** 10.3390/jof7110899

**Published:** 2021-10-25

**Authors:** Milan V. Kamble, Abhishek B. Shahapurkar, Shivakantkumar Adhikari, Nagaraja Geetha, Asad Syed, Bilal Ahmed, Sudisha Jogaiah

**Affiliations:** 1Laboratory of Plant Healthcare and Diagnostics, PG Department of Studies in Biotechnology and Microbiology, Karnatak University, Pavate Nagar, Dharwad 580003, Karnataka, India; milankamble115@gmail.com (M.V.K.); abhishekshahapurkar09@gmail.com (A.B.S.); adhikariklr93@gmail.com (S.A.); 2Nanobiotechnology Laboratory, Department of Studies in Biotechnology, University of Mysore, Mysore 570005, Karnataka, India; geethabiotech.uom@gmail.com; 3Department of Botany and Microbiology, College of Science, King Saud University, Riyadh 11451, Saudi Arabia; assyed@ksu.edu.sa; 4School of Chemical Engineering, Yeungnam University, Gyeongsan 38541, Korea; bilal22000858@yu.ac.kr

**Keywords:** grapevine, *Vitis vinifera*, downy mildew, *Plasmopara viticola*, microRNA, qRT-PCR

## Abstract

Downy mildew (DM) is one of the most devastating diseases disturbing viticulture, mainly during temperate and humid climates. The DM pathogen can attack grapevine leaves and berries differentially, and the disease is managed with recurring applications of fungicides that direct pathogen pressure, develop of resistant strains, and lead to residual soil toxicity and increased pollution effects. Plant microRNAs (miRNAs) are important candidates in physiological regulatory roles in response to biotic stress in plants. In this study, high-throughput sequencing and MiRDeep-P were employed to identify miRNAs in *Vitis vinifera*. Altogether, 22,492,910, 25,476,471, and 22,448,438 clean reads from the sterile distilled water (SDW)-control, bio-pesticide *Trichoderma harzianum* (TriH_JSB36)-treated, and downy mildew *Plasmopara viticola* pathogen libraries, respectively, were obtained. On the basis of the sequencing results and analysis (differential expression analysis), we observed significant differences in 15 miRNAs (5 novel upregulated, and 10 known downregulated) in the pathogen-infected sample (Test) in comparison to the SDW-control sample, with majority of the reads beingin the range of 20–24 bp. This study involves the identification and characterization of vvi-miRNAs that are involved in resistance against downy mildew disease in grapes.

## 1. Introduction

Grapevine (*Vitis vinifera* L.) is one of the most widely cultivated fruit crop with global economic significance. Wine production is greatly dependent on grapevine species and hence it is well acknowledged to mankind along with its mouth-watering juice. In India, the total cultivation of grapevine is about 34,000 hectares, and the total annual production is 1,000,000 tons per year [1]. The successful cultivation of grapevine suffers due to the attack of number of diseases, and hence appropriate plant management practices is of utmost importance in obtaining desired production levels. The disease-causing pathogen and the host encompass a dynamic relationship that can change with time. For the appropriate and feasible management, it is important to study the disease epidemiology and to understand the adaptation and variability of the DM pathogen to get a deeper insight into its life cycle.

The different varieties of *V. vinifera* cultivated worldwide are susceptible to various pathogens and are the reason for severe crop losses, including the oomycete *Plasmopara viticola*, the causal agent of the DM disease. Oomycetes are eukaryotic organisms with characteristics of fungus; they are unique that belong to the Stramenopiles, which also include the brown algae and diatoms. Oomycetes include terrestrial, freshwater, and marine organisms, but they are mainly pathogenic to plant and animals [2]. These pathogens cause many diseases in plants and animals causing huge impacts. In animals, Saprolegniosis mostly affects freshwater fishes, and the other notorious oomycete diseases include sudden oak death, white rust, potato late blight, and downy mildew in many plant species. Deeper studies related to the understanding of the oomycete pathogens virulence patterns and also the host adaptation strategies are the key to developing sustainable management practices [3].

Downy mildew disease is one of the most devastating diseases of grapes in viticulture areas with frequent rains, high humidity, and mild temperature in the summer period [4,5]. It is a common grapevine disease that has drawn much attention because of the severe economic losses it causes. The grape downy mildew pathogen *P. viticola* causes infection on the leaves, stem, and young berries. The pathogen forms oil spot lesions on the adaxial leaf surface along with severe sporulation on the abaxial leaf surface.

*P. viticola*, being an obligate oomycete biotroph, completes its whole life cycle on the host grapevine. Earlier, *P. viticola* was endemic only to North America, but in the 1870s it was accidentally introduced to Europe and caused epidemics on *V. vinifera* cultures all over Europe in the subsequent decades [5]. Downy mildew disease can be usually managed by chemical fungicide applications, such asmetalaxyl or Phosethyl-Al, neutralized Bordeaux mixture, and copper oxy-chloridemancozeb. However, indiscriminate use of fungicides not only leads to the emergence of resistant pathogens [6] but the impacts of pesticide overuse exert dangerous effects on human health and cause environmental pollution [7]. This concern has generated interest into the search for eco-friendly substitutes. Hence, research in the area of resistance mechanisms in grapes and development of sustainable management strategies for *P. viticola* infections is the current need.

In recent times, microRNAs (miRNAs) are found to exhibit important roles in controlling the devastating pathogen-causing disease in plants [8,9]. MicroRNAs (miRNAs) are endogenous ≈21 nt non-coding RNAs derived from single-stranded RNA precursors, and they form stem-loop structures [10,11]. MicroRNAs were first identified in *Caenorhabditis elegans* and eventually were observed in most of all eukaryotes [12]. miRNAs play a crucial role in higher plants in different developmental stages by regulating gene silencing at transcriptional and post-transcriptional levels [13,14,15]. Conventional methods do not generally provide information about these miRNAs. The next-generation sequencing (NGS) technology developed recently has been widely applied for genomic investigations such as genome sequencing and gene expression pattern analysis, along with small RNA sequencing. The advantage of ultra-high-throughput through this technology is that many new miRNAs with low abundance can be identified. NGS technologies generate an enormous amount of short sequence data, creating technical issues for bioinformaticists and computational biologists, as well as in analyzing NGS data in meaningful ways [16].

Plant miRNAs have significant roles in plant biotic stress responses. Plant miRNA genes are transcribed by RNA polymerase enzyme, which gives rise to miRNA (pri-miRNA). The pri-miRNA forms an unfit fold-back structure, further processed into a stem-loop precursor (pre-miRNA) by nuclear RNase-like enzymes known as DICER-LIKE proteins [17]. A mature miRNA sequence is made up of 19 to 24 nucleotides (nt) in length and is responsible for acting as a significant molecule in post-transcriptional gene silencing by base-pairing with target mRNAs. This leads to target mRNA fracture or translational repression, dependent on the degree of complementarily amongst the miRNA and its target transcript [10]. The same mature miRNA can be also present as several variants of their sequence in length. These members of miRNA variants are depicted as iso-miRNAs, the forms of microRNAs. During pre-miRNA processing, the iso-miRNAs are generated by alternative or indistinct cleavage of DICER [18]. Iso-miRNAs have been observed in both plants and animals [19].

A high-throughput degradome library sequencing technology has been developed for the global identification of targets of miRNAs in *Arabidopsis*, grapevine, and rice [20,21,22]. Many plant processes have recorded functions of plant miRNAs, which also includes developmental transitions [14,23], leaf growth [24], organ polarity [25], auxin signaling [26], and RNA metabolism [27,28,29]. Plants have evolved multiple orchestrated adaptive response mechanisms to re-program gene expression at the transcriptional, post-transcriptional, and post-translational levels to cope with biotic and abiotic stresses [30], and there are reports suggesting that there is increasing evidence indicated that miRNAs play important roles in plants in response to abiotic and biotic stresses [31].

For the determination of the transcriptional responses to downy mildew infection, profiling of transcripts has been successfully employed. Plant miRNAs were first described in *Arabidopsis thaliana* and were reported in other species eventually. The observation from different instances has emphasized that that these miRNAs play an important role in biotic and abiotic stress tolerance [30]. For example, in *Arabidopsis*, miR393 and other miRNAs are produced upon cold stress [29,32], and in rice plants, miR169g and miR393 are upregulated due to drought stress [33]. Under normal conditions, *Arabidopsis* miR398 directs the cleavage of *CSD1* and *CSD2* mRNA, bringing about downregulation of miR398 by inducing oxidative stress, which results in the production of *CDS1* and *CSD2* mRNAs [34]. Some investigations in *Arabidopsis* have shown that miR399, miR395, and miR398 are induced in response to phosphate-, sulfate-, and Cu^2+^-deprived conditions, respectively [15,31,35,36,37]. Moreover, among the 42 *Populus* miRNA families, the expression of some miRNAs is changed in response to abiotic stress conditions such ascold, heat, salinity, and dehydration, along with other mechanical stresses [38].

The miR393-guided post-transcriptional regulation plays an important role in the plant defense against pathogens targeting an auxin receptor, the transport inhibitor response 1 (*TIR1*) [39]. In an independent study, Goyal et al. [40] determined the role of miRNAs in defense signaling pathways by the involvement of NBS-LRR gene expression, production of reactive oxygen species (ROS), and hormone signals, which contribute to breed pathogen resistance plants. Transgenic *Arabidopsis* overexpressing miR393a exhibit increased tolerance to the pathogen *P. syringae* pv. tomato [30]. Another study from loblolly pine has also shown the expression of 10 miRNAs to be decreased in response to the rust fungus [41].

To detect new miRNAs participating in finding a way to create resistance against the downy mildew disease (DM)-causing pathogen, we constructed cDNA libraries to identify miRNAs involved in grapevine downy mildew resistance and validatedthe miRNA by qRT-PCR. To the best of our knowledge, this study is the first report to usehigh-throughput sequencing technology to investigate molecular events underlying induced disease resistance in grapes for downy mildew disease.

## 2. Materials and Methods

### 2.1. Plant Materials and Treatments

One-year-old grapevine plant cultivar Thompson was grown under field conditions. The pathogen, *Plasmopara viticola*, which was previously isolated from grapevine downy mildew-infected plants from Vijayapur, Karnataka, India, was used [4]. Young leaves from healthy plants were inoculated with the downy mildew pathogen *Plasmopara viticola*. Downy mildew-infected leaves selected from the crop field were collected in the evening hours, the harvested leaves were incubated overnight in moist conditions, sporangia-releasing zoospores were collected in the early morning using sterile distilled water (SDW), and the loads were adjusted to 5 × 10^−4^ zoospores mL^−1^ using a hemocytometer. Later, 10 randomly selected healthy grapevine plants were sprayed with grapevine downy mildew pathogen *P. viticola* suspension of 5 × 10^−4^ zoospores mL^−1^ (served as pathogen-infected plants); another set of 10 plants were smeared or sprayed with the bio-pesticide *Trichoderma harzianum* (TriH_JSB36) at 1 × 10^−8^ spores mL^−1^ untilrun-off (served as TriH_JSB-36-treated plants); and lastly, 10 plants were sprayed with sterile distilled water (served as SDW control plants) [42]. These three sets of treated plants (pathogen-infected, TriH_JSB36-treated, and SDW-control plants) were maintained in similar conditions. Sample leaves from the treated and control plants were collected in liquid nitrogen at 24 h after inoculation and were stored at −80 °C for further experiments [43].

### 2.2. Total RNA Isolation and Quality Assessment

Total RNA isolation was carried out from a frozen (24 h) three set of samples using the MagMAX Plant RNA Isolation Kit (Thermo Fisher Scientific, Waltham, MA, USA). Total RNA was isolated using the manufacturer’s protocol. The quality and quantity of total RNA was determined using Bioanalyzer and Qubit Fluorometer. Bioanalyzer uses a lab on a chip approach to perform capillary electrophoresis to analyze RNA. The technique is based on the use of a fluorescent dye that binds to RNA to determine RNA integrity. The fluorescent dye molecules intercalate into RNA strands. They are then detected by their fluorescence and translated into gel-like images (bands) and electropherograms (peaks). The integrity of RNA is determined by RNA integrity number (RIN) depending on the peaks. RIN values are measured from 1 to 10, where RIN value 1–5 indicates complete degradation, 5–7 indicates partially degraded RNA, and RIN values above 8 indicate good quality RNA.

The Qubit Fluorometer is a unique fluorometer designed to work seamlessly with highly specific and sensitive Qubit RNA quantification assays. The Quant-iT™ RiboGreen RNA Assay Kit contains Quant-iT™ RiboGreen RNA reagent along with buffers and RNA standards. Quant-iT™ RiboGreen RNA reagent is used as a very sensitive detection dye for the quantification of RNA in solution, with linear fluorescence detection in the range of 2–200 ng of RNA.

### 2.3. Small RNA Sequencing

Isolated RNA from treated and non-treated samples were used for small RNA high-throughput sequencing (HTS), and the sequence data were generated using Illumina HiSeq sequencing technology at the Clevergene Biocorp Pvt. Ltd., Bengaluru, India. The quality of the data was checked using FastQC [44] and MultiQC [45] software. The adapter sequences, low-quality bases, and the reads shorter than 17 bp were removed using the Trim Galore [46] tool.

### 2.4. Alignment and Identification of miRNA

The QC passed reads were mapped onto the indexed grapevine reference genome (GCF_000003745.3_12X); vine grape using the mapper.pl script of miRDeep2 [47]. Reads mapped to the reference genome were used to identify miRNAs with miRDeep2 using known and novel miRNA detection parameters. *V. vinifera* and *A. thaliana* (as related species) mature miRNAs from mirBase v22.1 [48] were used for miRNA prediction. A total of 389 novel and 76 known miRNAs were identified; miRNAs with miRDeep score less than 1 were excluded from further analysis, resulting in 284 novel and 63 known miRNAs. Expression levels of miRNAs were estimated using miRDeep2 quatifiler.pl script. For GO enrichment tests, we used Arabidopsis (TAIR10) peptide annotations, while BlastX program was used to align the unigenes with e-value ≤ 1 × 10^−5^. The top blast hits were deliberated as putative orthologous genes. *V. vinifera* unigenes were annotated with GO for Arabidopsis POGs. The GO was analyzed with BiNGO plugins [49] for Cytoscape using the hypergeometric test for statistical analysis. We used the Bonferroni correction method for e-value corrections to find over-representative terms with BiNGO.

### 2.5. Expression of Differential miRNAs

Differential expression analysis was carried out using the DESeq2 [50] package. The read counts were normalized, and differential expression was tested. MiRNAs with absolute log_2_ fold change ≥ 1, and *p*-value ≤ 0.05 was considered significant. A total of 15 miRNAs were significantly differentially expressed in pathogen-infected samples (Test) when compared to SDW-control samples. No significant miRNAs were identified in SDW-control vs. TriH_JSB36-treated and TriH_JSB36-treated vs. pathogen-infected comparisons. The expression profile of the differentially expressed miRNAs across the samples is presented in volcano plots and a heatmap.

## 3. Results

### 3.1. Small RNAs from Vitis vinifera

To identify downy mildew responsive miRNAs in grapevines, we generated total RNAs from young leaves (SDW-control, TriH_JSB36-treated, and pathogen-infected) for high-throughput sequencing, yielding 22,492,910, 25,476,471,and 22,448,438 raw read totals from the SDW-control, TriH_JSB36-treated, and pathogen-infected libraries, respectively (Table 1).

Following filtration, 13,460,335, 114,114,82, and 116,415,018 clean reads were obtained and mapped onto the *V. vinifera* reference genome, respectively (mature miRNAs from mirBase v22.1 and were used for miRNA prediction). The length distribution analysis of raw and filtered reads showed that the small RNA length from *V. vinifera* varied from 20 to 24 nt in both the SDW-control and the TriH_JSB36- treated and pathogen-infected groups (Figure 1).

### 3.2. miRNA Identification from Vitis vinifera

The processing of filtered reads from high-throughput sequencing was carried out according to miRDeep-P workflow, and the expression levels of miRNAs were estimated using miRDeep2 quatifiler.pl script with the *V. vinifera* genome as the reference sequence. For the SDW-control samples, a total of 347 miRNAs were identified, out of which 300 miRNAs were expressed, resulting in 63 known and 237 novel miRNAs. In the TriH_JSB36-treated sample, 321 expressed miRNAs including 63 known and 258 novel miRNAs were identified, and in the infected RNA samples, out of 304 expressed miRNAs, 63 known and 241 novel miRNAs were identified (Table 2).

On the basis of the sequencing results and analysis (differential expression analysis), we found significant differences in the expression study observed in 15 miRNAs expressed (five upregulated, and 10 downregulated) in the pathogen-infected sample (Test) in comparison to the SDW-control sample. The largest families obtained were vvi-miR395 family (vvi-miR395k, vvi-miR395i, vvi-miR395e, vvi-miR395d, vvi-miR395e, vvi-miR395e, vvi-miR395j, vvi-miR395a, vvi-miR395c and vvi-miR395l) with average −4.66 log_2_ fold change (downregulated). No significant miRNAs were identified in SDW-control vs. TriH_JSB36-treated and TriH_JSB36-treated vs. pathogen-infected comparisons (Table 3). Volcano plot and heatmap represented the expression profile of the differentially expressed miRNAs across the samples (Figure 2 and Figure 3, respectively).

### 3.3. Target Prediction of vvi-miRNAs

Targets for all significant novel and known miRNAs were obtained using miRanda [51], an algorithm for the detection of potential microRNA target sites in genomic sequences. The predicted targets were filtered on the basis of their pairing score (>150) and pairing energy score (<−20 Kcal/Mol). The identification and characterization of targets are essential to elucidate the functions of the miRNAs.

Finally, a total of 1061 known and 103 novel experimentally verified or putative transcripts were predicted for 347 vvi-miRNAs. The analysis showed that the many targets of conserved miRNA were transcription factors. Gene Ontology and pathways were enriched for target genes using DAVID [52]. Significantly enriched gene ontology terms and KEGG pathways are presented in Figure 4 for ‘regulation of biological process’, ‘signal transduction’, and ‘cellular response to stress’, which are the most enriched terms of biological process.

All the putative targets were analyzed for Gene Ontology (GO) and pathway analysis. In GO-categorized miRNAs, the three main GO terms were cellular component, KEGG for various miRNA involved, and molecular function. The significantly enriched terms of cellular component (Figure 5) included ‘regulation of cytoplasmic mRNA processing body assembly’, ‘nucleus’, and ‘intracellular membrane-bounded organelle’. 

Similarly, for miRNA involved in various pathways (Figure 6), the terms were spliceosome, phosphatidyl inositol signaling system, inositol phosphate metabolism, and RNA degradation.

In the molecular function (Figure 7), ‘DNA polymerase activity’ and ‘ATP-dependent NAD(P)H-hydrate dehydratase activity’ in catalytic activity, and ‘RNA binding’ and ‘protein complex’ were found to be significantly enriched terms (*p* < 0.05).

A total of 79 novel and 91 known miRNAs were identified on the basis of the previous studies on R genes along with GO terms. For novel miRNAs, 11 significant GO categories were enriched, and 9 of them belonged to cellular components. GO: 0044428 (nuclear part) was the most significant, reaching 5.39 × ^−4^, indicating that most of the genes were located in the nucleus, and 9 genes were further enriched in the nucleoplasm. In addition, 20 genes were enriched for GO: 0032991 (macroscopic complex). The other two significant categories belonged to biological processes, which are involved in mRNA metabolic and DNA packaging, respectively. For known miRNAs, there may be too few genes, and thus we did not find a significant category (FDR < 0.05). Therefore, we enlarged the scope of screening and selected the GO category with FDR < 0.08. A total of 21 genes are responsible in the regulating various biological processes, including seven genes involved in signal transduction (GO: 0050789), rhythm regulation, and stress response. In the molecular function category, 10 and 2 genes were enriched into RNA binding and DNA-directed DNA polymerase activity, respectively.

### 3.4. qRT-PCR Validation of miRNAs

To validate the accuracy and consistency of the micro-RNA-seq findings, we randomly selected 12 MIR genes to quantify the expression patterns using qRT-PCR analysis in SDW-control, TriH_JSB36-treated, and pathogen-infected grapevine leaf samples (Figure 8). In qRT-PCR analysis, four MIR genes showed higher activity in SDW-control plants, while they were slightly decreased in infected plants and significantly enhanced in the TriH_JSB36-treated plants. Three genes depicted normal expression in SDW-control but were moderately enhanced in pathogen-infected and TriH_JSB36-treated plants. However, the remaining five MIR genes showed enhanced activity at a very low level in treated grapevine leaf samples. Overall, the qRT-CPR findings were consistent with high-throughput sequencing data, reflecting the accuracy of the experimental findings.

## 4. Discussion

The complex miRNA-mediated regulatory networks controlling various physiological processes requires in depth studies that will lay the path to unravel identifying the entire set of miRNAs and their targets. Several genetic studies have reported that organisms may contain about 1% miRNA genes of the total protein-coding genes [53,54,55]. These miRNAs are widely present in animals and plants as the key regulators of these gene expressions [56,57,58]. There are 243 and 511 miRNAs annotated in *Arabidopsis* and rice, respectively, according to the miRBase database [59,60]. For resistance breeding in the commercially cultivated *V. vinifera* to be enhanced, the identification of major gene regulators during downy mildew susceptibility and resistance is crucial. In the present study, total RNAs from young leaves (sterile distilled water (SDW)-control, biopesticide, TriH_JSB36-treated, and pathogen-infected) were extracted for high-throughput sequencing, yielding 22,492,910, 25,476,471,and 22,448,438 raw read totals from the SDW-control, TriH_JSB36-treated, and pathogen-infected libraries, respectively. The primary goal of this study was the identification of *P. viticola* pathogenicity factors involved in the infection process of grapevine. To enable the use of transcriptomic approaches, we first sequenced the RNA isolated from the infected plants followed by assembling the *P. viticola* genome.

For the screening of miRNA, the most widely used method is small RNA high-throughput sequencing, encouraging the detection of an enormous number of sRNA in different plant species. Hence, the present study also took up this method to identify miRNAs in *V. vinifera*. In most of the research studies, the common lengths of plant sRNA are usually 21 or 24 nt [56,61,62]. According to reports, in some plants, the number of 24 ntsRNA usually exceeds 21 nt sRNA, for example, in gymnosperms *Taxus chinensis* [63];in monocots *Oryza sativa* [64]; and in eudicots *Arabidopsis* [56], tomato [62], and *Citrus trifoliate* [16]. Similarly in the present study, the length distribution analysis for row and filtered reads showed that the small RNA length from *V. vinifera* varied from 20 to 24 nt in both the SDW-control and the TriH_JSB36 and pathogen-treated groups.

In this study, the sequence data were generated using Illumina HiSeq sequencing technology, and a total of 347 miRNAs for SDW-control samples were identified, out of which 300 miRNAs were expressed, resulting in 63 known and 237 novel miRNAs. In the TriH_JSB36-treated sample, 321 expressed miRNAs including 63 known and 258 novel miRNAs were identified, and in infected RNA samples, out of 304 expressed miRNAs, 63 known and 241 novel miRNAs were identified. Similar supporting data were documented by Yang et al. [65], wherein the authors identified the role of isolated miR482 involved in modulating resistance against *Verticillium dahlia* in potato, with the resistance being correlated with the suppression of NBS-LRR genes. The same trend in disease resistance to *Fusarium oxysporum* was observed in tomato with miR482 [66].

The relative expression between TriH_JSB36-treated vs. SDW-control plant miRNAs and the respective target genes in the trans-regulating state was higher than in the cis-regulating state. Nevertheless, the expression profiles of some miRNAs were negatively related to their target genes (Figure 3). Here, in this study, we found that the significant expression of the 15 miRNAs, of which 5 miRNAs were upregulated in control conditions. Our results are in agreement with rice miRNA (miR7695) that upregulated resistance against *M. oryzae* [67].

Further, we studied the target genes for the entire significant novel and known miRNAs that were obtained using miRanda. For the plant miRNA identification, MiRDeep-P is an efficient and efficacious tool. In this study, small RNA (sRNA) high-throughput sequencing and data analysis by MiRDeep-P not only has detected known vvi-miRNAs, but also has predicted 258 novel miRNAs from the grapevine leaves treated with fungicides. Previous studies also have reported several novel miRNAs from the *V. vinifera* grapevine cultivars [68,69]. Wang et al. reported novel miRNAs from the grapevine cultivar Summer Black [70]. In another study, an increased innate immune response of potato to *Phytophthora infestans* was observed by auxin response target genes [71]. Although, these novel miRNAs were reported in the above studies, they were not found in this study. The reason could be the experimental conditions used for library preparations and as well as the geographical and environmental factors of the cultivars.

A fascinating finding from the present study is that the number of target genes that have been selected for miRNA expression is very large. On the basis of the previous studies, we found that most of the expressed miRNAs (conserved) and their targets are transcription factors. miRNAs play a crucial role in plants by regulating growth and development and are responsible for survival [72]. However, the function of conserved miRNAs is not always conserved [73], e.g., for each of the three conserved miRNA families, miR478a, miR473a, and miR482, the miRNA families from rice and *Populus* have assorted functions. The conserved miRNAs families are primarily involved in stress responses, although the non-conserved miRNAs targets are significantly varied, unfolding the diverse roles of miRNAs in different biological systems.

## 5. Conclusions

This study explores the molecular defense mechanisms of grapes to downy mildew.At the level of post-transcription regulation, we identified miRNAs from a highly resistant grapevine cultivar. Combined with sRNA high-throughput sequencing, bioinformatics, and molecular biology technologies, the identified vvi-miRNAs further need to be investigated for their role in downy mildew resistance to *V. vinifera*. Moreover, in-depth studies on these miRNAs and their target genes will further expand our understanding of the molecular mechanisms underlying downy mildew resistance in *V. vinifera*, paving a way to detect the possible involvement of NBS-LRR gene expression in defense signals as in other cases.

## Figures and Tables

**Figure 1 jof-07-00899-f001:**
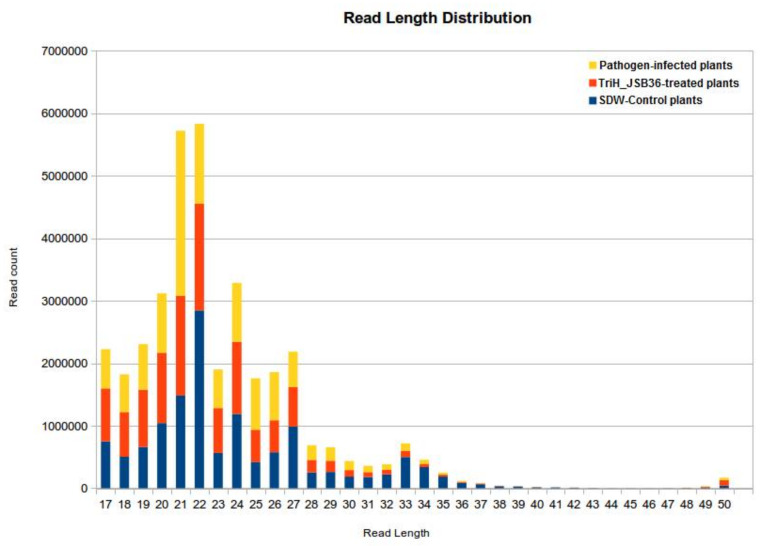
Length distribution analysis of reads for SDW-control, TriH_JSB36-treated, and pathogen-infected (Test) samples used for miRDeep2 analysis.

**Figure 2 jof-07-00899-f002:**
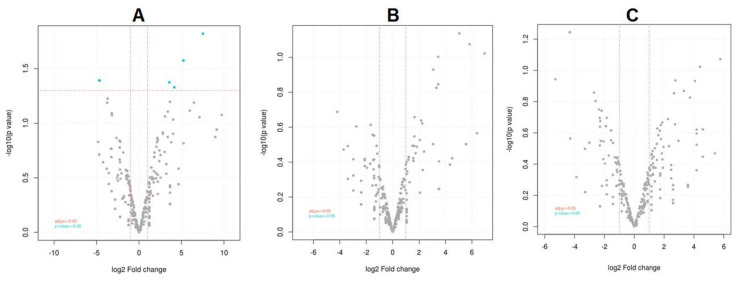
Volcano plot showing differential expression level of miRNAs. (**A**) Volcano plot for SDW-control vs. pathogen-infected plants, (**B**) SDW-control vs. TriH_JSB36-treated plants, (**C**) TriH_JSB36-treated vs. pathogen-infected plants.

**Figure 3 jof-07-00899-f003:**
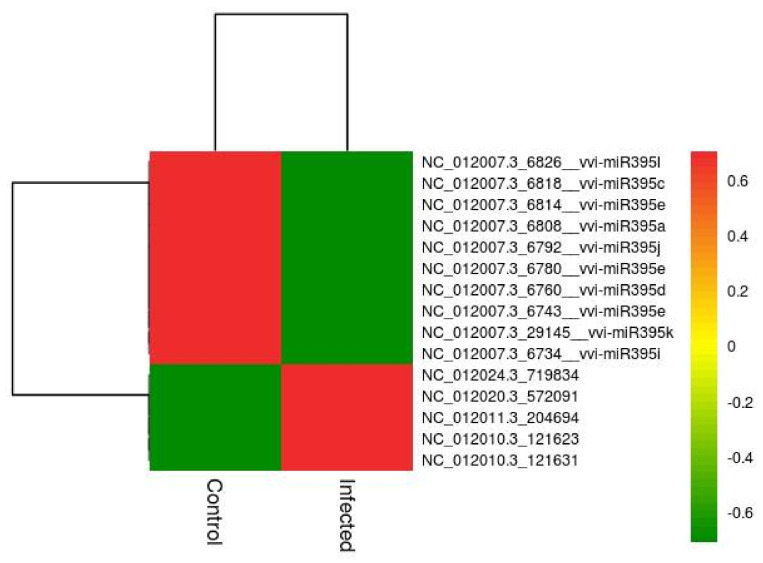
Heatmap of expression profile (Log_2_-based values) (upregulation vs. downregulation) of the differential expressed miRNAs in SDW-control vs. pathogen-infected samples. The heatmap was generated using the Rstudio program (A package of R, USA).

**Figure 4 jof-07-00899-f004:**
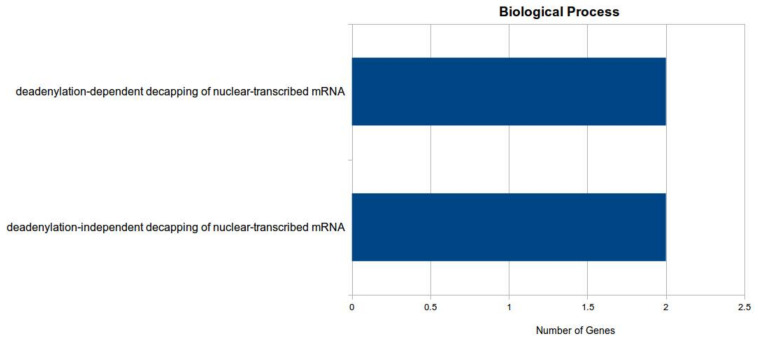
Gene Ontology annotation for all the predicted targets of miRNA genes involved in deadenylation-dependent and independent decapping of nuclear transcribed mRNA.

**Figure 5 jof-07-00899-f005:**
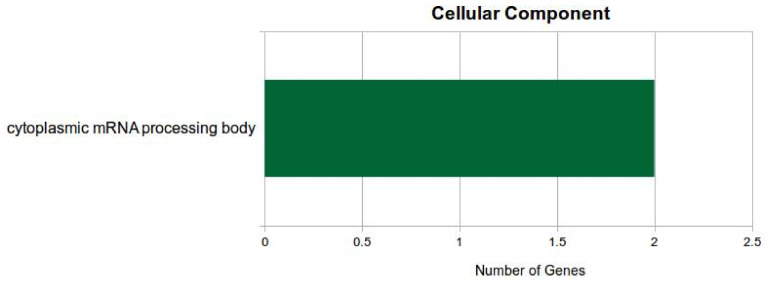
Number of differentially expressed miRNA genes in cytoplasmic mRNA processing body.

**Figure 6 jof-07-00899-f006:**
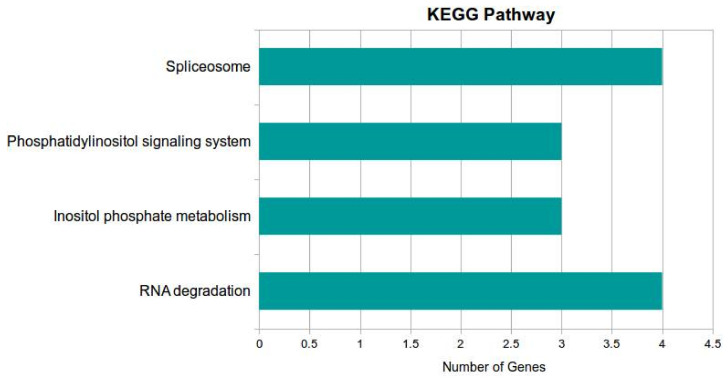
miRNAs involved in various pathways, namely, spliceosome, phosphatidyl inositol signaling system, inositol phosphate metabolism, and RNA degradation.

**Figure 7 jof-07-00899-f007:**
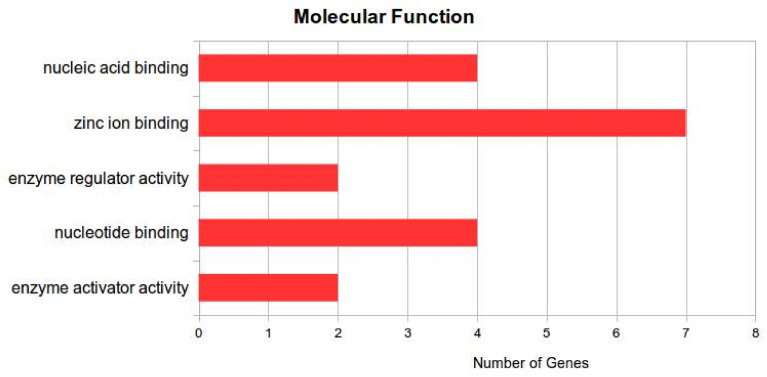
Genes involved in nucleic acid binding, zinc ion binding, enzyme regulator activity, nucleotide binding, and enzyme activator activity.

**Figure 8 jof-07-00899-f008:**
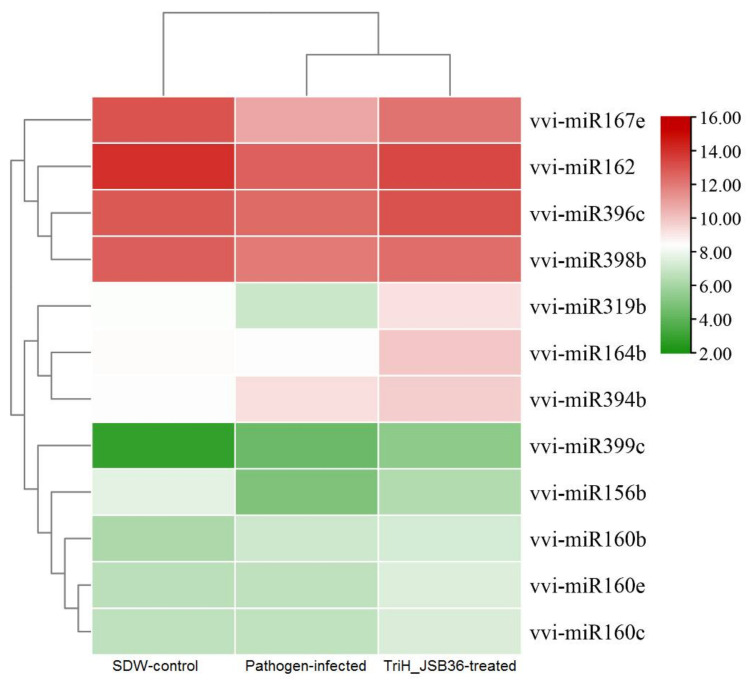
Expression of target mRNA transcript by qRT-PCR in response to SDW-control, pathogen-infected, and TriH_JSB36-treated plants. The heatmap was developed using Log_2_ value from the Rstudio program (A package of R, USA).

**Table 1 jof-07-00899-t001:** Number of raw reads and read length distribution before and after QC filtering and trimming of SDW-control, TriH_JSB36-treated, and pathogen-infected (Test) samples used for miRDeep2 analysis obtained by high-throughput sequencing.

Raw Reads		
Sample Name	Number of Reads	Read Length
SDW-control	22,492,910	50
TriH_JSB36-treated	25,476,471	50
Pathogen-infected (Test)	22,448,438	50
**Reads after QC Filtering and Trimming**		
**Sample Name**	**Number of Reads**	**Read Length**
SDW-control	13,460,335	17–50
TriH_JSB36-treated	11,411,482	17–50
Pathogen-infected (Test)	11,641,501	17–50

**Table 2 jof-07-00899-t002:** Identification of total novel and known miRNAs obtained from SDW-control, TriH_JSB36-treated, and pathogen-infected (Test) samples with miRDeep score.

Sample Name	Total miRNAs	Expressed miRNAs	Known miRNAs	Novel miRNAs
SDW-control	347	300	63	237
TriH_JSB36-treated	347	321	63	258
Pathogen-infected (Test)	347	304	63	241

**Table 3 jof-07-00899-t003:** Number of differentially expressed (i.e., upregulated and downregulated) miRNAs obtained from SDW-control vs. pathogen-infected, SDW-control vs. TriH_JSB36-treated, and TriH_JSB36-treated vs. pathogen-infected (Test) samples.

Condition (Control vs. Test)	Significantly Expressed miRNAs	Upregulated miRNAs	Downregulated miRNAs
SDW-control vs. pathogen-infected	15	5	10
SDW-control vs. TriH_JSB36–treated	0	0	0
TriH_JSB36-treated vs. pathogen-infected	0	0	0

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
