# Peer review of "Identification and Characterization of Downy Mildew-Responsive microRNAs in Indian Vitis vinifera by High-Throughput Sequencing"

_jof, 2021, doi:10.3390/jof7110899_

Round 1

Reviewer 1 Report

The work is well done, some parts are a bit confused or not very clear; authors need to rewrite some parts of the chapter material and methods (marked in the pdf) and improve disscussion. 

Author Response

Responses to the Reviewers’ Comments

The authors would like to thank the Reviewers for his/her constructive comments and suggestions that have helped us improve our manuscript. An extensive revision has been undertaken and incorporated all the corrections and suggestions raised by the Reviewers in the revised manuscript.

Reviewer #1: The work is well done, some parts are a bit confused or not very clear; authors need to rewrite some parts of the chapter material and methods (marked in the pdf) and improve disscussion. 

Response: We are very glad that the Reviewer highly evaluated our manuscript, our special thanks to this Reviewer for providing the edited PDF. We have edited the manuscript based on your suggestions mentioned in the PDF copy and we truly believe that your suggestions and comments have helped us improve the overall quality of our manuscript.

Reviewer 2 Report

The authors performed high throughput sequencing technology to identify miRNAs that are involved during downy mildew disease in Vitis vinifera. Because this work is the first report in this topic, the study is relevant. However, some issues may be attended to improve the manuscript.

Introduction:

Please correct all the typos and some unclear sentences through it. For example:

Lane 126: Another study fromloblolly pine has, please correct from (space) loblolly

Lane 25 and 26: there were significant differences the expression were observed in 15 miRNAs, please rephrase the sentence

Lane 60: The plant parts infected by P. viticola are leaves, young berries. Please rewrite

Lane 64: In recent times, microRNAs (miRNAs) are found to exhibit important roles in controlling the devastating pathogen causing disease in plants. Please include references.

I suggest to the authors incorporate information about  the defense response at the molecular level in this pathosystem. There is information in this regard, for example: 10.1016 / j.ygeno.2019.02.011.

Finally, in this section I  suggest to re organize substantially the information included along lanes 74 to 126. Would be better if first describe the biosynthesis and regulation of micro RNAs, follow by the evidences of the microRNA during pathogenesis and finally the technology  used. Because now the information are mixed and unclear.

Materials and methods:

Lane 136: please include information about P.viticola (origin)

Results

I recommend to the authors include photographs of the infected plants. At what time the symptoms are observed?

Lane 195: total RNAs from young leaves (Control, fungicide treated, and infected) were generated. Please clarify if the fungicide treated is the Trichoderma treatment that you mention in materials and methods. If this is true I think the term fungicide is not correct since the definition of fungicide is “ any substance used to kill or inhibit the growth of fungi”. So I suggest to rename and maybe in the discussion section justify the use of Trichoderma strain.

The results shown in figures 4 to 7 can be included in a single figure

Discussion and conclusions.

I kindly suggest that the authors improve the both sections.I can not identify the potential role of the miRNAs in the biotic stress described. This suggestion is to be congruent in your statement “This study involves the identification and characterization of vvi-miRNAs that are involved in resistance against downy mildew disease in grapes”.

Author Response

Responses to the Reviewers’ Comments

The authors would like to thank the Reviewers for his/her constructive comments and suggestions that have helped us improve our manuscript. An extensive revision has been undertaken and incorporated all the corrections and suggestions raised by the Reviewers in the revised manuscript.

Reviewer #2: The authors performed high throughput sequencing technology to identify miRNAs that are involved during downy mildew disease in Vitis vinifera. Because this work is the first report in this topic, the study is relevant. However, some issues may be attended to improve the manuscript.

Response: We would like to express our special thanks to the Reviewer for evaluating our manuscript positively.  Also, thanks for providing constructive comments that have helped us improve the quality of our manuscript. We have put in all efforts to revise the manuscript, taking into account all the comments and suggestions of the Reviewer.

Introduction:

Comment 1: Please correct all the typos and some unclear sentences through it. For example:

Lane 126: Another study fromloblolly pine has, please correct from (space) loblolly

Response: Thank you so much for this critical observation. We have corrected the typos errors. Also, we have carefully fixed all the typos/topographical errors in the revised manuscript.  

Comment 2: Lane 25 and 26: there were significant differences the expression were observed in 15 miRNAs, please rephrase the sentence

Response: We thank this Reviewer for this comment. The sentence is now re-written as follows:

“…..significant differences were observed in 15 miRNAs (5 novel up-regulated, and 10 known down-regulated) in the pathogen-infected sample ….”

Comment 3: Lane 60: The plant parts infected by P. viticola are leaves, young berries. Please rewrite

Response: We thank the Reviewer for this critical observation. To meet the Reviewer comment, the sentence has been revised as follows:  

“The grape downy mildew pathogen, P. viticola causes infection on the leaves, stem and young berries”

Comment 4: Lane 64: In recent times, microRNAs (miRNAs) are found to exhibit important roles in controlling the devastating pathogen causing disease in plants. Please include references.

Response: Thank you very much for this comment and suggestion which we totally agree. The following citations have been included in the revised manuscript:

[8] Djami-Tchatchou, A.T.; Sanan-Mishra, N.; Ntushelo, K.; Dubery, I.A. Functional Roles of microRNAs in Agronomically Important Plants-Potential as Targets for Crop Improvement and Protection. Front. Plant Sci. 2017, 8, 378. doi: 10.3389/fpls.2017.00378”

[9] “Yang, X.; Zhang, L.; Yang, Y.; Schmid, M.; Wang, Y. miRNA Mediated Regulation and Interaction between Plants and Pathogens. Int. J. Mol. Sci. 2021, 22, 2913. https://
doi.org/10.3390/ijms22062913”.

Comment 5: I suggest to the authors incorporate information about  the defense response at the molecular level in this pathosystem. There is information in this regard, for example: 10.1016 / j.ygeno.2019.02.011.

Response: We thank the Reviewer for this important suggestion. Accordingly, we have now mentioned the role of miRNAs in plant defense response against biotic stress. The following sentence has been incorporated along with the reference:

“In an independent study, Goyal et al. [40], determined the role of miRNAs in defense signaling pathways by the involvement of NBS-LRR gene expression, production of reactive oxygen species (ROS), and hormone signals, which contributes to breed pathogen-resistance plants“.

[40] “Goyal, N.; Bhatia, G.; Sharma, S.; Garewal, N.; Upadhyay, A.; Upadhyay, S.K.; Singh, K. Genome-wide characterization revealed role of NBS-LRR genes during powdery mildew infection in Vitis vinifera. Genomics. 2020, 112(1), pp. 312-322. doi: 10.1016/j.ygeno.2019.02.011”.

Comment 6: Finally, in this section I suggest to re organize substantially the information included along lanes 74 to 126. Would be better if first describe the biosynthesis and regulation of micro RNAs, follow by the evidences of the microRNA during pathogenesis and finally the technology used. Because now the information are mixed and unclear.

Response: We highly appreciate this Reviewer for this comment which we totally agree. To meet the Reviewer comment, with the available few supporting literatures, we have put all our efforts to revise both discussion with appropriate citations and also the conclusion section.

Materials and methods:

Comment 7: Lane 136: please include information about P.viticola (origin)

Response: Thank you so much for this comment. The source of the pathogen is now incorporated in the revised manuscript.  

Results

Comment 8: I recommend to the authors include photographs of the infected plants. At what time the symptoms are observed?

Response: An image showing the expression of downy mildew symptoms is now included in the revised manuscript. The disease was observed during early December, 2019.

Comment 9: Lane 195: total RNAs from young leaves (Control, fungicide treated, and infected) were generated. Please clarify if the fungicide treated is the Trichoderma treatment that you mention in materials and methods. If this is true I think the term fungicide is not correct since the definition of fungicide is “ any substance used to kill or inhibit the growth of fungi”. So I suggest to rename and maybe in the discussion section justify the use of Trichoderma strain.

Response: We thank the Reviewer for this critical observation of our manuscript. Yes, the Reviewer has rightly pointed out, it is Trichoderma harzianum (TriH_JSB36)-treated plants that was by mistake mentioned as fungicide. We have now replaced the usage of the word fungicide with TriH_JSB36 in the whole manuscript.

Comment 10: The results shown in figures 4 to 7 can be included in a single figure

Response: We are very sorry to say that it is difficult to combine the figures 4-7, since the parameters of the figures 4-7 are completely different and we would like to supply the figures separately so that we can reach wide audience.

Discussion and conclusions.

Comment 11: I kindly suggest that the authors improve the both sections. I cannot identify the potential role of the miRNAs in the biotic stress described. This suggestion is to be congruent in your statement “This study involves the identification and characterization of vvi-miRNAs that are involved in resistance against downy mildew disease in grapes”.

Response: Thank you so much for this comment and suggestion. Since, the role of identified miRNAs is not fully described for their involvement in resistance against grapevine downy mildew pathosystem. The discussion and conclusion section is revised based on the below facts:

“The identified vvi-miRNAs further needs to be investigated for its role in downy mildew resistance to V. vitifera. Moreover, indepth studies on these miRNAs and their target genes will further expand our understanding of the molecular mechanisms underlying downy mildew resistance in V. vitisfera will pave a way to detect the possible involvement of NBS-LRR gene expression in defense signals as in other cases”.